# Effects of Crawling before Walking: Network Interactions and Longitudinal Associations in 7-Year-Old Children

**DOI:** 10.3390/ijerph19095561

**Published:** 2022-05-03

**Authors:** Jorge Cazorla-González, Sergi García-Retortillo, Mariano Gacto-Sánchez, Gerard Muñoz-Castro, Juan Serrano-Ferrer, Blanca Román-Viñas, Abel López-Bermejo, Raquel Font-Lladó, Anna Prats-Puig

**Affiliations:** 1University School of Health and Sport (EUSES), University of Girona, 17003 Girona, Spain; jcazorla@euses.cat (J.C.-G.); sgarciaretortillo@gmail.com (S.G.-R.); marianogacto@um.es (M.G.-S.); gmunoz@euses.cat (G.M.-C.); jserrano@euses.cat (J.S.-F.); broman@euses.cat (B.R.-V.); rfont@euses.cat (R.F.-L.); 2Keck Laboratory for Network Physiology, Department of Physics, Boston University, Boston, MA 02215, USA; 3Bronchiectasis Group, Girona Biomedical Research Institute, Dr. Josep Trueta Hospital, 17007 Girona, Spain; 4CIBER Fisiopatología de la Obesidad y Nutrición (CIBERobn), Instituto de Salud Carlos III (ISCIII), 28029 Madrid, Spain; 5Pediatric Endocrinology Group, Girona Biomedical Research Institute, 17003 Girona, Spain; alopezbermejo@idibgi.org; 6Department of Pediatrics, Dr. Josep Trueta Hospital, 17003 Girona, Spain; 7Department of Medical Sciences, University of Girona, 17003 Girona, Spain; 8Research Group of Culture and Education, Institute of Educational Research, University of Girona, 17003 Girona, Spain; 9Research Group of Clinical Anatomy, Embryology and Neuroscience (NEOMA), Department of Medical Sciences, University of Girona, 17003 Girona, Spain

**Keywords:** crawling, children, cardiometabolic

## Abstract

Background: To study the impact of crawling before walking (CBW) on network interactions among body composition, the cardiovascular system, lung function, motor competence and physical fitness, at age 7, and to assess the longitudinal association between CBW and body composition, the cardiovascular system, lung function, motor competence, physical fitness and physical activity parameters, at age 7. Method: CBW, body composition, cardiovascular system, lung function, motor competence, physical fitness and physical activity were assessed in seventy-seven healthy Caucasian children. Results: Network analyses revealed that the crawling group had a greater number of links among all the studied variables compared with the non-crawling group. In the longitudinal study, using multiple regression analyses, crawling was independently associated with fat mass (%), fat-to-muscle ratio and systolic blood pressure, with models explaining up to 56.3%, 56.7% and 29.9% of their variance, respectively. Conclusions: CBW during child’s development is a possible modulator in the network interactions between body systems and it could influence future metabolic and cardiovascular health.

## 1. Introduction

Hands and knees crawling before walking is considered the first skill that provides children with independent mobility. It is a rhythmic movement between the upper and lower limbs characterized by diagonal coordination [1], arm strength and balance [2], which is essential for adaption to the environment and to enhance the development of general motor skills [3,4]. As for other motor skills, changes in crawling postures and proficiency are related to improvement in locomotor proficiency and changes in infants’ brains [5]. Children who are more motivated to achieve independent mobility tend to have early gross motor development, such as sitting up, crawling and walking; this may help to prevent later obesity [6] and provides great benefits in body composition, cardiovascular health, lung function and physical fitness [7,8]. Conversely, delayed motor-milestone achievement at 9 months of age has been associated with increased sedentary time and lower levels of physical activity in later childhood [9]. These beneficial effects could be partly driven by an improvement in muscle–muscle and organ–organ interactions [10]. 

Different physiological systems are communicated through levels and time scales that lead to changes in the connectivity, complexity and diversity of the functional organ interaction within networks, promoting good health [11]. As pointed out by Bartsch et al. [12], understanding the nature of such interactions can provide useful information regarding the specific role of several physiological systems (e.g., musculoskeletal, cardiovascular, and respiratory) within an integrated network.

Body composition has been extensively related to cardiovascular health and it is known that raised blood pressure during childhood represents an increased risk of cardiovascular diseases in adulthood [13,14]. Interestingly, fat-to-muscle ratio (FMR) has recently been pointed out as a simple and useful indicator in the assessment of the risk for metabolic syndrome [15] and cardiovascular diseases [16] in adults. The FMR considers both the accumulation of fat mass and skeletal muscle mass. The negative effects of an excessive fat-mass accumulation on blood pressure are well known, but scarce data exist regarding the protective role of skeletal muscle on blood pressure [17]. We therefore hypothesize that crawling before walking will modify interactions among body composition, cardiovascular system, lung function, motor competence and physical fitness, at age 7. In addition, crawling before walking will be related to the same body system parameters in healthy children.

To the best of the author’s knowledge, there are no studies exploring the impact of the absence of hands and knees crawling before walking on network interactions and longitudinal associations in healthy children. Thus, the purposes of our study were to assess: (1) the impact of crawling before walking on network interactions among body composition, cardiovascular system, lung function, motor competence and physical fitness, at age 7, and (2) the longitudinal association between crawling before walking with body composition, cardiovascular system, lung function, motor competence, physical fitness and physical activity, at age 7.

## 2. Materials and Methods

### 2.1. Study Population

A total of 103 children were invited to participate in the study, of whom 26 were excluded (not meeting inclusion criteria, *n* = 4; their parents declined to participate, *n* = 16; no crawling information provided, *n* = 6; Appendix A). A total of 77 apparently healthy school-age children (37 boys and 40 girls; age 7.49 ± 0.34 yr) were finally included in a longitudinal retrospective case-control study. Subjects were recruited from 5 schools. All subjects were of Caucasian origin.

The inclusion criteria were: (1) children between 7 and 8 years of age, in their second year of primary school; and (2) apparently healthy children. Exclusion criteria were: (1) major congenital abnormalities; (2) evidence of chronic illness or chronic use of medication; (3) musculoskeletal or neurological disease; (4) motor functional limitations; (5) pain or dysfunction in the upper or lower extremities; (6) cardiovascular abnormalities or hypertension; and (7) missing crawling data.

### 2.2. Sample Size Estimation

To determine the sample size for this study, a power analysis was conducted using G*Power 3.1. Previous research assessing crawling effects on visual prediction abilities in spatial object processing have reported large effect sizes (Cohen’s d = 0.59). In this previous study, 9-month-old infants’ visual prediction abilities in the context of spatial object processing were doubled in crawlers when compared with non-crawling infants [18]. Thus, using an effect size of d = 0.59, α < 0.05, power (1−β) = 0.80, we estimated a total sample size of 74. 

### 2.3. Measures

Measurements were assessed by the same expert observer who was unaware of the crawling characteristics of the participants. Body composition, cardiovascular system, lung function, motor competence and physical fitness data were collected at schools. Physical activity data were collected with triaxial accelerometers for seven consecutive days (24 h/day) and a crawling questionnaire was given to the parents in a specific information session at the beginning of the project.

#### 2.3.1. Body Composition

Weight was measured wearing light clothes with a calibrated scale (Portable TANITA; 240MA, Amsterdam, The Netherlands) in kilograms, and height was measured using a wall-mounted stadiometer (SECA SE206, Hamburg, Germany) in centimeters. Body mass index (BMI) for each participant was calculated using the formula (weight divided by the square of height in meters). BMI z-scores were standardized according to age- and sex-adjusted values from regional normative data [19]. Waist circumference was measured in the standing position at the umbilical level in centimeters. Body composition (fat and muscle mass) was assessed by bioelectric impedance (Portable TANITA; 240MA, Amsterdam, The Netherlands), collected under fasting conditions and with an empty bladder. Fat-to-muscle mass ratio (FMR) was calculated by dividing fat mass by muscle mass, both in kilograms [20].

#### 2.3.2. Cardiovascular System

Blood pressure (BP), in mmHg, was measured in the supine position on the right arm after 5 minutes rest; a validated sphygmomanometer (OMRON M3 Intellisense, Kyoto, Japan), with a cuff size appropriate for arm circumference was used. Averages of three readings were recorded for each subject. 

#### 2.3.3. Lung Function

Forced vital capacity (FVC) and forced expiratory volume in 1 s (FEV_1_), both in milliliters, were obtained by performing forced spirometry (In2itive Vitalograph, Lenexa, KS, USA) in accordance with international standards guidelines in children [21].

#### 2.3.4. Motor Competence

Locomotor skills (LS), object control skills (OCS) and motor competence (MC) were examined by the Canadian Agility and Movement Skill Assessment (CAMSA) test [22]. The final score was obtained as CAMSA points, the maximum score was 14 and children who were able to accurately combine the speed and skill components of the assessment obtained the highest score. A detailed explanation of the methodology used to assess MC in our study has been previously published [23].

#### 2.3.5. Physical Fitness

Cardiorespiratory fitness (CRF) was assessed by means of a half-mile run test. The objective of this test was to complete an 800-meter track, around 2 cones 40 m apart, in the quickest possible time. The total time taken to run half a mile was recorded in minutes. A shorter time reflected higher cardiorespiratory fitness.

Lower-body muscular strength (LBMS) was assessed, in centimeters, using the long jump test. Each child was requested to jump as far as possible, with their feet together and while remaining in the upright posture. The best result from two attempts was used in the subsequent analysis.

Upper-body muscular strength (UBMS) was measured, in kilograms, using a handgrip strength test. It consisted of squeezing an analog dynamometer (TKK 5001, Grip-A, Takei, Tokyo) gradually and continuously for at least 5 s. Grip span was standardized at 5.0 cm. The test was performed twice for each hand, recording the highest value for each hand. The average of these two results was used in the analysis as a measure of upper-body muscular strength.

#### 2.3.6. Physical Activity

Moderate to Vigorous Physical Activity (MVPA) was measured with triaxial accelerometers (ActiGraph GT3X, Actigraph Corporation, Pensacola, FL, USA) placed around children’s waists for seven consecutive days (24 h/day) and expressed in minutes of MVPA per day. Accelerometers were programed in epochs of one second and data were analyzed using ActiLife software (ActiGraph LLC, Pensacola, Florida, USA). The Evenson cut-off points were used to define PA intensities (sedentary (0 to 100 counts per minute), moderate intensity (2296–4011 counts per minute) and vigorous intensity (4012 or more counts per minute)) [24]. Periods of 20 min of consecutive zero counts were selected as non-wear time and removed from the analysis. Only those registers with at least 4 days of at least 10 h of valid recording time per day, including 1 weekend day, were considered for the analysis. The mean valid time dedicated to moderate and vigorous physical activity was used for the present study.

#### 2.3.7. Crawling Assessment

Crawling was defined as self-produced locomotion using hands and knees in infants, which took place before walking and during the infant’s gross motor development [25]. Parents of all children included in the study were asked to fill in a questionnaire which asked if their child crawled before walking (YES/NO). We made sure that all parents understood that YES meant that their child used their hands and legs to support their displacement when their back was straight and that this must have occurred before their child began walking [18,25].

### 2.4. Study Groups

To analyze the possible longitudinal effects of crawling before walking on body composition, cardiovascular system, lung function, motor competence and physical fitness and activity, the total sample (*n* = 77) was split into two groups according to crawling data (non-crawling (*n* = 35) and crawling before walking (*n* = 42)).

### 2.5. Correlation Matrices and Networks

A set of different variables related to the following categories was selected: body composition, cardiovascular system, lung function, motor competence, and physical fitness and activity (Table 1). To study the interactions among such variables, one correlation matrix and one network was used, for both crawling and non-crawling groups. To create the correlation matrix (Figure 1A), the Pearson correlation coefficient was used to calculate the correlations between the different pairs of variables. Subsequently, the corresponding network (Figure 1B) was obtained by using only the statistically significant correlations obtained in the correlation matrix. With the aim of quantifying the interactions within each network, we also computed the total number of links occurring for each variable (i.e., the number of significant correlations) and the percentage of difference, in both the non-crawling and the crawling groups (Figure 1C). As shown in Figure 1A,B, the links were divided into six types: strong positive links (SPL; Pearson coefficients > 0.8), intermediate positive links (IPL; 0.6 < Pearson coefficients < 0.8), weak positive links (WPL; 0.32 < Pearson coefficients < 0.6), weak negative links (WNL; −0.32 > Pearson coefficients > −0.6), intermediate negative links (INL; −0.6 > Pearson coefficients > −0.8) and strong negative links (SNL; Pearson coefficients < −0.8). The visualization framework used in this study was based on previous studies analyzing network interactions among physiological systems during different physiological states [13,26]. The correlation matrices and networks were processed and obtained by means of Matlab R2016b (Mathworks, Natik, MA, USA).

### 2.6. Data Analysis

SPSS version 22.0 (SPSS Inc, Chicago, IL, USA) was used to perform data analysis. All data were tested for normality by using a Kolgomorov–Smirnov test. Differences between groups were analyzed using a Student’s *t*-test. The relationships between variables were analyzed by Pearson correlation followed by multiple regression analysis using the enter method. Significance was set at *p* < 0.05.

## 3. Results

### 3.1. Study Population

A sample of 77 children (mean age = 7.49 ± 0.34 years; 52% female) was assessed. Table 1 shows the results for clinical assessments, body composition, cardiovascular system, lung function, motor competence, physical fitness and physical activity parameters for all the studied children. Results are also presented based on crawling group.

### 3.2. Network Interactions: The Impact of Crawling before Walking on the Body Systems at Age 7

When analyzing interactions between body system parameters (Figure 1), the crawling group showed a higher number of links between all pair categories, compared with the non-crawling group (Figure 1B). The variables with a higher number of links with the percentage of difference between groups were, in the non-crawling group: fat mass, FMR and LS (100%) and CRF (14.28%), and in the crawling group: FEV_1_, OCS and LBMS (100%), UBMS (66.66%), waist (60%), muscle mass and MC (50%), SBP (42.85%) and FVC (9.09%) (Figure 1C).

### 3.3. Longitudinal Association between Crawling before Walking and the Studied Variables

In children who crawled before walking, decreased values (compared with children who did not crawl during infancy) were observed for the following clinical assessments: BMI z-score (decreased by 114% (*p* = 0.022)), body composition (fat mass percentage by 14.4% (*p* = 0.009) and FMR by 18.5% (*p* = 0.008)) and the cardiovascular system (SBP decreased by 7% and DBP by 10% (both, *p* = 0.001)) (Appendix A and Table 1). No differences were observed for weight and height z-scores, waist, muscle mass, lung function (FVC and FEV_1_), motor competence (LS, OCS and MC), physical fitness variables (CRF, LBMS and UBMS) and MVPA per day.

In a multiple regression analysis with fat mass (%) as a dependent variable, crawling, sex and weight z-score were independently associated with fat mass (%), explaining 56.3% of its variance. In a multiple regression analysis with FMR as a dependent variable, crawling, sex and weight z-score were independently associated with the FMR, explaining 56.7% of its variance. In a multiple regression analysis with SBP as a dependent variable, crawling, MVPA per day and the FMR were independently associated with SBP, explaining 29.9% of its variance (Table 2).

## 4. Discussion

In our network analysis, children who crawled before walking showed a higher number of interactions among variables assessing body composition, cardiovascular system, lung function, motor competence and physical fitness and activity, in comparison with children who did not crawl. Moreover, in our longitudinal results, previous crawlers had lower fat mass, higher levels of muscle mass compared with fat mass and lower systolic blood pressure. Crawling before walking during a child’s development could be considered a possible modulator in the interaction of networks between body systems and could influence anthropometric and cardiovascular parameters at 7 years of age. According to the literature, 7 years of age is the first period of adiposity rebound and could be important for predicting BMI in young adults [27,28].

A previous study assessing the age at crawling showed no associations between crawling onset and any anthropometric outcome; the authors suggested that BMI may be a poor marker of adiposity in early childhood [6]. The results of the current study, may suggest that a gross motor stimulus at early stages of life, such as crawling, positively impacts on children’s health, increasing the links between different variables of body composition, cardiovascular system, lung function, motor competence and physical fitness, and diminishing fat mass (%), FMR and blood pressure levels in childhood. The difference between the results obtained could be explained by the fact that the absence of crawling before walking was the focus of the study, rather than the age of crawling. There was also more emphasis placed on the number of network interactions, fat mass (%), FMR and SBP, rather than on the BMI at 7 years of age. 

Physical activity can increase the physiological interaction between systems [12] and could alter the skeletal muscle and adipose tissue phenotype. This promotes an increase in energy expenditure, reducing fat mass [29,30] and its potential adipogenic progenitor cells [31]. During the first year of infancy, crawling reflects habitual physical activity and regular movement [2]. Since crawling can appear as early as in the fourth month of life [32], providing opportunities for fostering crawling (or at least for not hampering it) may be the most effective way to increase early interaction between the variables of body composition, cardiovascular system, lung function, motor competence and physical fitness, and in turn can be beneficial for overall health during childhood. According to Bartsch et al. [13], the coordinated interactions of organ systems are essential in maintaining health and generating different physiological states. Thus, crawling can challenge and disrupt muscle homeostasis and lead to muscle tissue structural and functional adaptations, as it is a repeated stimulus continuing for a certain amount of time [33]. Previous studies have reported that skeletal muscle can retain molecular information, so is primed for future plasticity following encounters with the same stimuli. Furthermore, epigenetic changes could be the mechanistic explanation for this muscle memory [34]. The present study would suggest that crawling may act as a repeated stimulus that can program skeletal muscle memory, improving the balance of skeletal muscle and adipose tissue from infancy. Skeletal muscle and adipose tissue are often closely related to the same phenotype or outcomes. Both tissues secrete molecules capable of modulating local and systemic metabolism. Myokines and adipokines secreted from the corresponding tissues have an important effect in maintaining a balanced ratio of skeletal fat-to-muscle mass, and thus, may play a key role in the modulation of body composition and metabolism [17]. This study hypothesizes that crawling can lead to skeletal muscle and adipose tissue adaptations that increase the interaction between both tissues, which in turn will improve cardiovascular health, and may also promote an anti-inflammatory phenotype resulting from a diminished FMR and fat mass percentage. This is not surprising since early life factors such as birth weight and even fetal growth are related to muscle size, strength, and hypertension in later life [35]. However, more studies are needed to elucidate the mechanisms whereby crawling can regulate skeletal muscle mass and adipose tissue, and thus contribute to shaping an interaction network of early body systems which assist in preventing obesity and high blood pressure as early as childhood.

High systolic blood pressure is defined as one of the five leading risks of death worldwide [36]. The results of this study show that children who did not crawl during infancy have higher levels of systolic and diastolic blood pressure at school age. This could be due to the association of SBP with levels of adipose and muscle tissue during childhood [37]. This observation is in line with previous studies showing that early infant locomotive development (lifting the body upwards to a standing position, walking with support and walking without support) was an early life predictor of adult blood pressure levels [38]. Studies from animal models suggested that locomotion and blood pressure are closely connected through shared regulatory systems [39]. Many of the risk factors for cardiovascular diseases are modifiable, and early childhood may be a critical period for the prevention of excess adiposity and low muscle mass. Efficacious and cost-effective interventions are needed to alter determinants of abnormal body composition [40]. Crawling can be enhanced by stimulating touch, gravity, and movement receptors to develop body schema, motor-planning abilities and proprioceptive functioning. Moreover, adding challenges that involve unexpected forces applied from multiple planes of movement can improve necessary pelvic and trunk control for crawling [41]. However, more studies are needed to ascertain if fostering crawling during infancy could improve metabolic and cardiovascular health later in life.

The results of this study should be interpreted in the light of several limitations. First as a longitudinal retrospective study, there was a reliance on parental recall regarding children’s crawling. Second, during the study, no real-time recordings of the variables were collected. Third, the study was limited to a population in the same province, and therefore needs to be replicated in other populations. Future studies with real-time recordings and larger samples sizes would help analyze the potential effect of interaction between systems and additional confounding variables.

### Implications for Practice

The absence of crawling before walking could have a negative impact on the interactions between systems, fat-to-muscle ratio, and systolic blood pressure at age 7. Pediatric physical therapists could provide opportunities which advocate for the non-hampering of crawling before walking, which may be an effective way to improve the correlation between system variables that contribute to overall future health, especially metabolic and cardiovascular health during childhood.

## 5. Conclusions

The results portray crawling before walking during a child’s development as a possible modulator in the interaction of networks between body systems at 7 years of age. Additionally, crawling before walking could have a possible effect on anthropometric and cardiovascular parameters at 7 years of age. These results suggest that crawling before walking could have a positive impact on the future overall health of participants.

## Figures and Tables

**Figure 1 ijerph-19-05561-f001:**
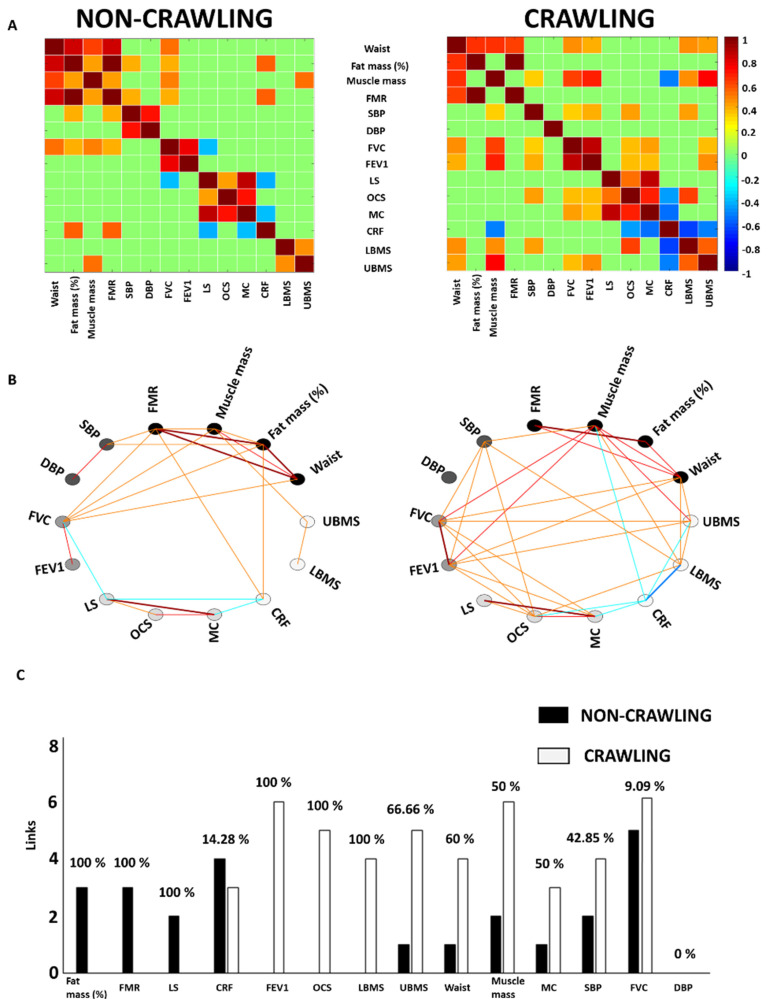
Network interactions change between body systems depending on whether children crawled or not before walking. (**A**) Body systems matrix. The matrices show the Pearson correlation coefficient between body systems. Non-significant correlations are represented in green. (**B**) Body systems networks. Each node represents a specific body system for each category (body composition (dark black), cardiovascular system (dark gray), lung function (gray), motor competence (light grey) and physical fitness (bone white)). Links between two nodes represent the coupling strength (i.e., the Pearson coefficient) between two genes. The links are divided into six types: strong positive links (SPL; Pearson coefficients > 0.8), intermediate positive links (IPL; 0.6 < Pearson coefficients < 0.8), weak positive links (WPL; 0.32 < Pearson coefficients < 0.6), weak negative links (WNL; −0.32 > Pearson coefficients > −0.6), intermediate negative links (INL; −0.6 > Pearson coefficients > −0.8), and strong negative links (SNL; Pearson coefficients < −0.8). (**C**) Total number of linked interactions for each variable. The height of the bar corresponds to the number of significant correlations in the body systems matrix.

**Table 1 ijerph-19-05561-t001:** Clinical assessments, body composition, cardiovascular system, lung function, motor competence and physical fitness variables in the study subjects (*n* = 77) by crawling group.

Variable	Non-Crawling	Crawling	*p*-Value
**Clinical assessments**			
N	35	42	
Age (y)	7.53 ± 0.36	7.46 ± 0.32	0.365
Sex (% female)	54	50	
Weight z-score	−0.27 ± 0.76	−0.52 ± 0.58	0.110
Height z-score	−0.05 ± 1.05	−0.07 ± 0.98	0.944
BMI z-score	−0.27 ± 0.66	−0.58 ± 0.48	0.022
**Body composition**			
Waist (cm)	56.8 ± 5.20	55.4 ± 3.90	0.184
Fat mass (%)	20.1 ± 5.30	17.2 ± 3.91	0.009
Muscle mass (kg)	19.5 ± 2.36	19.0 ± 2.38	0.410
Fat to muscle ratio (kg)	0.27 ± 0.09	0.22 ± 0.06	0.008
**Cardiovascular system**			
SBP (mmHg)	104 ± 10	97 ± 7	0.001
DBP (mmHg)	62 ± 9	56 ± 6	0.001
**Lung function**			
FVC (mL)	1672.81 ± 261.9	1621.57 ± 255.31	0.413
FEV1 (mL)	1474.33 ± 264.44	1453.51 ± 250.86	0.744
**Motor competence**			
Locomotion skills (CAMSA points)	0.29 ± 0.10	0.27 ± 0.11	0.540
Object control skills (CAMSA points)	0.14 ± 0.07	0.14 ± 0.06	0.820
Motor competence (CAMSA points)	14.7 ± 4.21	14.56 ± 4.16	0.882
**Physical fitness**			
Cardiorespiratory fitness (min)	5.23 ± 0.62	4.94 ± 0.77	0.130
Lower-body muscular strength (cm)	91.58 ± 14.11	94.88 ± 13.98	0.393
Upper-body muscular strength (kg)	20.86 ± 3.89	19.67 ± 6.25	0.329
**Physical Activity**			
MVPA per day (min/day)	60.33 ± 20.16	62.28 ± 24.03	0.718

BMI: body mass index; SBP: systolic blood pressure; DBP: diastolic blood pressure; FVC: forced vital capacity; FEV1: forced expiratory volume in 1 s; MVPA: moderate to vigorous physical activity. Data are expressed as mean ± SD. *p*-values are from Student’s *t*-tests.

**Table 2 ijerph-19-05561-t002:** Multivariate linear models in all children (*n* = 77; crawlers (*n* = 42) and non-crawlers (*n* = 35)). Upper table: fat mass (%) as a dependent variable, middle table: fat-to-muscle ratio as dependent variable and lower table: systolic blood pressure as dependent variable.

		Beta	Sig.	R²
**Fat Mass (%)**				
	Crawling	−0.185	0.038	
	Age (y)	0.048	0.575	
	Sex	−0.178	0.031	
	MVPA per day	0.018	0.223	
	Weight z-score	0.687	<0.0001	
				0.563
**Fat-to-Muscle Ratio**				
	Crawling	−0.188	0.024	
	Age (y)	0.437	0.664	
	Sex	−0.198	0.017	
	MVPA per day	−0.085	0.312	
	Weight z-score	0.689	<0.0001	
				0.567
**Systolic Blood Pressure**				
	Crawling	−0.316	0.005	
	Age (y)	0.053	0.622	
	Sex	0.137	0.190	
	MVPA per day	0.290	0.008	
	Fat-to-muscle ratio	0.305	0.008	
				0.299

## Data Availability

Data are available upon request.

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
