# Peer review of "Effects of Crawling before Walking: Network Interactions and Longitudinal Associations in 7-Year-Old Children"

_ijerph, 2022, doi:10.3390/ijerph19095561_

Round 1
Reviewer 1 Report
This study evaluated the effects of CBW on body composition, cardiovascular system, lung function, motor ability and physical activity of 7-year-old children, found the positive effects of CBW on FMR and Systolic blood Pressure, and put forward suggestions for children to crawl before walking, which can provide beneficial reference for promoting children's health. However, the following problems still exist in this study:
1. Is "in seven 7-year-old children" correct?
2.Line 49: The first appearance of FMR requires full name;
3. Crawling evaluation is the core content of this study, but the reliability of the evaluation method of this study is not high, which is reflected in the following two aspects :(1) although the researcher has mentioned recall bias in the limitation, it is still a key factor affecting the reliability of the conclusions of this study. (2) Crawling evaluation is not detailed enough. Crawling requires the accumulation of a certain amount to have an impact on variables such as body composition, and the description of this study does not provide substantial advice on crawling information.
Author Response
Response to Reviewer 1 Comments
Comments and Suggestions for Authors: This study evaluated the effects of CBW on body composition, cardiovascular system, lung function, motor ability and physical activity of 7-year-old children, found the positive effects of CBW on FMR and Systolic blood Pressure, and put forward suggestions for children to crawl before walking, which can provide beneficial reference for promoting children's health. However, the following problems still exist in this study:
We appreciate the comments received, with them we have been able to make improvements and increase the quality of the document. We would like to clarify the queries received:
Point 1: Is "in seven 7-year-old children" correct?
Response 1: Thanks for the appreciation. For this study, we included children from the second year of primary school, and they were aged between 7 and 8. This has been amended properly in the manuscript.
Point 2: Line 49: The first appearance of FMR requires full name.
Response 2: The error has been identified and the document has been modified.
Point 3: Crawling evaluation is the core content of this study, but the reliability of the evaluation method of this study is not high, which is reflected in the following two aspects :(1) although the researcher has mentioned recall bias in the limitation, it is still a key factor affecting the reliability of the conclusions of this study. (2) Crawling evaluation is not detailed enough. Crawling requires the accumulation of a certain amount to have an impact on variables such as body composition, and the description of this study does not provide substantial advice on crawling information.
Response 3: Thanks for this comment, we agree that this must be well clarified in the document. We have used the concept of hands-and-knees crawl according to the study by Bodnarchuk JL & Eaton WO [25]. Parents were explained that the crawling stage is before walking and that the infants can only use their hands and legs as support for displacement and straight back. In our study, we clearly defined crawling to all families to be sure that they all used the same criteria when answering the questionnaire. Those families who were in doubt have been excluded (as indicated in the document).
Please see now the following information added to the main document:
“We made sure that all parents understood that YES meant that their child used their hands and legs to support their displacement when their back was straight and that this must occur before their child began walking [25]”.
- Bodnarchuk JL, Eaton WO. Can parent reports be trusted?: Validity of daily checklists of gross motor milestone attainment. Applied Developmental Psychology. 2004;25(4),481–490. doi:10.1016/j.appdev.2004.06.005

Reviewer 2 Report
First of all, I would like to congratulate the authors for the clarity and specificity of the paper. I think the research has a great practical application. In general the document is very clear and explains step by step the procedure followed. However, I would like to detail some aspects that could increase the quality of the document.
In the summary, it is recommended to include numerical values for some of the findings in the results section.
The introductory section is a bit poor, there are previous studies, which say about the subject and which have a different study from them. The motivation for the study should be justified a bit more.
Before 2.3. there should be a section called "measures".
From section 2.3. onwards, in the part of the different measures, the unit of measurement of most of the variables should be indicated, as many of them are not included, some of them are specified in the subsequent comments.
Line 98, include weight measurement instrument. And units of measurement for weight, height and circumference.
Line 102, what are the normative data you mention, reference.
Line 125, the CAMSA assesses time and from a scoring rubric, so what score or how was it assessed, specify.
It is recommended not to put text next to the figure and thus enlarge the figure, as the first images do not have sufficient resolution.
In the results section perhaps table one should go first, as it is a descriptive table of everything that will follow.
There are differences in BMI, % fat mass, cardiovascular variables, against those who did not crawl.
Table 1, the units of measurement for the variables "motor competence" and "physical fitness" are missing.
In table 2, why were variables included that were not significant? This could affect the precision of the estimation.
Why do they justify everything at the age of 7, if they include in their inclusion criteria up to the age of 9 years? Significant growth and maturational changes occur from age 6 to 9, so this wide range could affect your results. Please justify this or put it as a limitation of the study.
Author Response
Response to Reviewer 2 Comments
Comments and Suggestions for Authors: First of all, I would like to congratulate the authors for the clarity and specificity of the paper. I think the research has a great practical application. In general the document is very clear and explains step by step the procedure followed. However, I would like to detail some aspects that could increase the quality of the document.
We appreciate the comments received, with them we have been able to make improvements and increase the quality of the document. We would like to clarify the queries received:
Point 1: In the summary, it is recommended to include numerical values for some of the findings in the results section.
Response 1: Some numerical data has been added to the summary, in the results section.
Point 2: The introductory section is a bit poor, there are previous studies, which say about the subject and which have a different study from them. The motivation for the study should be justified a bit more.
Response 2: Thanks for the appreciation. Changes have been made in the original document to improve the justification of the need for the study of crawling and the absence of crawling before walking in infants.
Point 3: Before 2.3. there should be a section called "measures".
Response 3: The section ''2.3. Measures’’ have been added and the variables that have been collected appear now as subsections.
Point 4: From section 2.3. onwards, in the part of the different measures, the unit of measurement of most of the variables should be indicated, as many of them are not included, some of them are specified in the subsequent comments.
Response 4: Thanks, the units of measurement of the variables have been added.
Point 5: Line 98, include weight measurement instrument. And units of measurement for weight, height and circumference.
Response 5: The measurement units and the measuring instrument for weight have been added.
Point 6: Line 102, what are the normative data you mention, reference.
Response 6: The reference used has been added in the manuscript (19. Puente M, Canela J, Alvarez J, Salleras L, Vicens-Calvet E. Cross-sectional growth study of the child and adolescent population of Catalonia (Spain). Ann Hum Biol. 2009;24:435-452. doi:10.1080/03014469700005202).
Point 7: Line 125, the CAMSA assesses time and from a scoring rubric, so what score or how was it assessed, specify.
Response 7: Thanks for raising this, we have added some extra information and references to clarify the CAMSA assessment.
Point 8: It is recommended not to put text next to the figure and thus enlarge the figure, as the first images do not have sufficient resolution.
Response 8: Thank you, this point has been amended accordingly.
Point 9: In the results section perhaps table one should go first, as it is a descriptive table of everything that will follow.
Response 9: The proposed change has been made.
Point 10: There are differences in BMI, % fat mass, cardiovascular variables, against those who did not crawl.
Response 10: Thanks for the comment. We agree with the reviewer about the findings. However, we were more focused on adipositiy variables (such as fat mass % and FMR) and systolic blood pressure rather than BMI and dyastolic blood pressure since previous articles have shown that BMI may be a poor marker of adiposity in early childhood [6] and high systolic blood pressure is defined as one of the five leading risks of death worldwide [34]. We were also interested in FMR because we wanted to highlight the importance of fat mass and muscle mass.
- Benjamin Neelon SE, Oken E, Taveras EM, Rifas-Shiman SL, Gillman MW. Age of achievement of gross motor milestones in infancy and adiposity at age 3 years. Matern Child Health J. 2012;16(5):1015-1020. doi:10.1007/s10995-011-0828-3
- GBD 2017 Risk Factor Collaborators. Global, regional, and national comparative risk assessment of 84 behavioural, envi-ronmental and occupational, and metabolic risks or clusters of risks for 195 countries and territories, 1990-2017: a systematic analysis for the Global Burden of Disease Study 2017. Lancet Lond Engl. 2018;392(10159):1923-1994. doi:10.1016/S0140-6736(18)32225-6
Point 11: Table 1, the units of measurement for the variables "motor competence" and "physical fitness" are missing.
Response 11: Thanks. Changes have been made accordingly.
Point 12: In table 2, why were variables included that were not significant? This could affect the precision of the estimation.
Response 12: We completely agree with the reviewer but what we want to show in the article is that the association between crawling and fat mass percentage and FMR is independent of physical activity. Thus, we decided to include all non-significant variables in the final model. However, we were sure that this did not affect the precision of the estimation. We want to show the reviewer here the same model with just the significant variables. As it can be seen we obtain almost the same results with and without the non-significant variables included in the model.
|
|
|
Beta |
Sig. |
R² |
|
Fat mass (%) |
|
|
|
|
|
|
Crawling |
-0.172 |
0.034 |
|
|
|
Sex |
-0.193 |
0.016 |
|
|
|
Weight z-score |
0.672 |
<0.0001 |
|
|
|
|
|
|
0.550 |
|
|
|
|
|
|
|
Fat-to-muscle ratio |
|
|
|
|
|
|
Crawling |
-0.169 |
0.037 |
|
|
|
Sex |
-0.212 |
0.008 |
|
|
|
Weight z-score |
0.674 |
<0.0001 |
|
|
|
|
|
|
0.557 |
|
|
|
|
|
|
|
Systolic blood pressure |
|
|
|
|
|
|
Crawling |
-0.324 |
0.004 |
|
|
|
MVPA per day |
0.290 |
0.003 |
|
|
|
Fat to muscle ratio |
0.315 |
0.009 |
|
|
|
|
|
|
0.331 |
Point 13: Why do they justify everything at the age of 7, if they include in their inclusion criteria up to the age of 9 years? Significant growth and maturational changes occur from age 6 to 9, so this wide range could affect your results. Please justify this or put it as a limitation of the study.
Response 13: Thanks for pointing that out. It was a mistake. For this study we included children from the second year of primary school, and they were aged between 7 and 8. This has been amended in the methods sections.

Round 2
Reviewer 1 Report
Review of manuscript entitled " Effects of crawling before walking: Network interactions and longitudinal associations in 7-year-old children".
The revised manuscript is sufficiently well elaborated and scientifically structured.
The methodology have been carefully determined.
The results have been presented in a very concise and clear manner.
Overall, the manuscript is well written and interesting.
I have several minor concerns.
- Why choose a 7-year-old child, whether this age is representative or for other reasons, is not explained in the article.
- Line 92 "from the second year of primary school" or " second grade children "?
In view of the foregoing and the review, I consider that the article should be published with this small modification.
Author Response
Response to Reviewer 1 Comments
Comments and Suggestions for Authors: Review of manuscript entitled " Effects of crawling before walking: Network interactions and longitudinal associations in 7-year-old children".
The revised manuscript is sufficiently well elaborated and scientifically structured.
The methodology have been carefully determined.
The results have been presented in a very concise and clear manner.
Overall, the manuscript is well written and interesting.
I have several minor concerns.
In view of the foregoing and the review, I consider that the article should be published with this small modification.
First of all, thank you for your comments and suggestions that allowed us to greatly improve the quality of the manuscript.
Point 1: Why choose a 7-year-old child, whether this age is representative or for other reasons, is not explained in the article.
Response 1: Thanks for raising this, we have added some extra information and references to clarify the reasons for choosing 7-year-olds children: ‘’According to the literature, 7 years of age is the first period of adiposity rebound and could be important for predicting BMI in young adults [27,28].’’
- Kang MJ. The adiposity rebound in the 21st century children: meaning for what?. Korean J Pediatr. 2018;61(12):375-380. doi:10.3345/kjp.2018.07227.
- Williams S, Davie G, Lam F. Predicting BMI in young adults from childhood data using two approaches to modelling adiposity rebound. Int J Obes Relat Metab Disord. 1999 Apr;23(4):348-54. doi: 10.1038/sj.ijo.0800824.
Point 2: Line 92 "from the second year of primary school" or " second grade children "?
Response 2: Thanks for the appreciation. The error has been identified and the document has been modified.
